# Whole-Head Noninvasive Brain Signal Measurement System with High Temporal and Spatial Resolution Using Static Magnetic Field Bias to the Brain

**DOI:** 10.3390/bioengineering11090917

**Published:** 2024-09-13

**Authors:** Osamu Hiwaki

**Affiliations:** Graduate School of Information Sciences, Hiroshima City University, 3-4-1 Ozuka-Higashi, Asa-Minami-Ku, Hiroshima 731-3194, Japan; hiwaki@hiroshima-cu.ac.jp

**Keywords:** noninvasive brain measurement, magnetically biased field, movement-related brain signal, brain–machine interface

## Abstract

Noninvasive brain signal measurement techniques are crucial for understanding human brain function and brain–machine interface applications. Conventionally, noninvasive brain signal measurement techniques, such as electroencephalography, magnetoencephalography, functional magnetic resonance imaging, and near-infrared spectroscopy, have been developed. However, currently, there is no practical noninvasive technique to measure brain function with high temporal and spatial resolution using one instrument. We developed a novel noninvasive brain signal measurement technique with high temporal and spatial resolution by biasing a static magnetic field emitted from a coil on the head to the brain. In this study, we applied this technique to develop a groundbreaking system for noninvasive whole-head brain function measurement with high spatiotemporal resolution across the entire head. We validated this system by measuring movement-related brain signals evoked by a right index finger extension movement and demonstrated that the proposed system can measure the dynamic activity of brain regions involved in finger movement with high spatiotemporal accuracy over the whole brain.

## 1. Introduction

Measuring brain signals is fundamental for understanding complex cognitive processes, neurological diseases, and applications for brain–machine interfaces. To this end, noninvasive brain signal measurement techniques are safer alternatives to invasive methods and support both research and clinical applications. Conventional noninvasive methods to measure brain signals include electroencephalography (EEG), magnetoencephalography (MEG), functional magnetic resonance imaging (fMRI), and near-infrared spectroscopy (NIRS). EEG and MEG, in particular, have high temporal resolution [1]. EEG measures the electrical activity of the brain using electrodes placed on the scalp [2,3], and its primary advantage lies in its high temporal resolution because it can measure brain activity on the order of milliseconds. This makes EEG particularly useful for studying dynamic cognitive processes and detecting abnormalities such as epilepsy [4]. However, EEG has limited spatial resolution because the recorded signals are affected by the nonuniformity of electrical conductivity in the head [5]. Thus, it is difficult to accurately locate the source of brain activity using EEG [6]. On the other hand, MEG captures the magnetic fields generated by neuronal activity using highly sensitive magnetometers, such as superconducting quantum interference devices (SQUIDs) [7,8], resulting in better spatial resolution compared to EEG, as magnetic fields are less distorted by the nonuniformity of the electrical properties of the head [9]. Nevertheless, similar to EEG, MEG suffers from the inverse problem with no unique solution. Multiple electrical sources in the brain can produce identical magnetic field patterns; therefore, it is difficult to pinpoint the exact origin of brain activity [10,11]. fMRI measures brain activity by detecting changes in blood flow based on the principle that neuronal activity is coupled with regional blood oxygenation [12]. This technique offers high spatial resolution, allowing for detailed mapping of brain areas involved in specific cognitive tasks. However, fMRI has lower temporal resolution than EEG and MEG, as the hemodynamic response occurs over seconds [13,14]. NIRS uses near-infrared light to measure changes in blood oxygenation and brain volume through optical fibers placed on the scalp [15,16]. The data obtained from NIRS are indirect signals of neural activity based on hemodynamic responses, potentially complicating data interpretation because changes in blood oxygenation do not always correlate directly with neural activity. NIRS has a lower spatial resolution than fMRI [17,18]. Despite these advances in noninvasive techniques for measuring brain signals, optimizing temporal and spatial resolution in a single instrument remains challenging [19]. This requirement is critical for a comprehensive understanding of the dynamic processes in the human brain. Brain–machine interfaces (BMI), which use brain signals to operate computers and devices, require dynamic and spatially accurate brain information. However, the conventional techniques to measure brain function do not provide sufficient information for BMI [20].

We developed a novel noninvasive brain signal measurement technique with high temporal and spatial resolution using a method that biases a static magnetic field emitted from a coil on the head to the brain [21]. In this study, we introduced a groundbreaking system for whole-head noninvasive measurement of brain signals with excellent temporal and spatial resolution using this technique. The system was validated by measuring movement-related signals evoked by volitional finger movements.

## 2. Materials and Methods

### 2.1. Noninvasive Brain Signal Measurement System Using Static Magnetic Field Bias to the Brain

The proposed system biases a static magnetic field emitted from a coil to the brain for the noninvasive measurement of localized dynamic brain signals (Figure 1) [21]. The static magnetic field emitted from the coil located on the scalp reaches the cerebral cortex, which is the outermost layer of the brain. The static magnetic field fluctuates by neuroelectric activity as it passes through the cerebral cortex and returns to the top of the coil. A magnetic field fluctuating with the electrical activity of the cerebral cortex can be obtained by a magnetic sensor placed at the top of the coil. Therefore, this technique enables the measurements of localized neural activity in the cerebral cortex beneath the coil, which consists of a 0.32 mm diameter enamel copper wire wound tightly around a 5 mm diameter, 25 mm high permalloy cylinder to generate a static magnetic field. The coil is placed in a plastic cylinder with a screw thread that can be coupled with a plastic nut. We used a highly sensitive magnetic sensor (MI-CB-1DH-M-B, Aichi Steel, Tokai, Aichi, Japan) that can measure magnetic field fluctuation in the order of nanotesla. A plastic nut was attached to the edge of the magnetic sensor with resin along the sensor axis. The magnetic sensor and coil were connected by joining the nut to the screw thread.

We developed a whole-head brain signal measurement system consisting of 159 pairs of magnetic sensors and coils. The coils were mounted on an elastic neoprene cap such that each longitudinal axis of the coil and magnetic sensor were perpendicular to the scalp (Figure 2). The coils were connected in series, and a direct current was supplied from a power supply. The current in the coils was adjusted using a potentiometer. During the measurements, we maintained a direct electrical current in the coils to produce a magnetic field of approximately 40 µT at the bottom of each coil. The signal outputs of the magnetic sensors were acquired using data acquisition modules (NI-9205, National Instruments, Austin, TX, USA) inserted into slots in a chassis (cDAQ-9179, National Instruments, Austin, TX, USA). The signals from the 159 magnetic sensors were simultaneously sent to the PC through the USB port of the cDAQ-9179 and acquired by a system developed by LabVIEW (National Instruments, Austin, TX, USA).

### 2.2. Verification of the System by Measurement of Movement-Related Signal

Three right-handed healthy participants (one female and two males) were included in this study. This study was approved by the institutional review board of Hiroshima City University (18-2) and conducted in accordance with the principles expressed in the Declaration of Helsinki. Written informed consent was obtained prior to the experiments.

The experiment was conducted in a magnetically shielded room. The right hand of the participant was placed on a wooden platform on a table, where the participant performed a simple motor task involving the extension of the right index finger. The participant was instructed to extend the right index finger as quickly and naturally as possible. The interval between adjacent trials was voluntarily maintained at approximately 5 s. The initiation of finger movement was detected by an infrared beam sensor installed on the contact surface between the platform and fingertip, and a trigger signal from the infrared beam sensor was transferred to the data acquisition system. The signals were measured from 2000 ms before to 1000 ms after the finger movement. The movement-related signals were obtained by averaging 50 trials per session. The signals were averaged, and topographic maps were drawn using MATLAB (MathWorks, Natick, MA, USA).

## 3. Results and Discussion

The movement-related signals evoked by volitional finger movements were successfully obtained (Figure 3). The largest and most localized signal was observed at the channel indicated as (A) in Figure 3; it was located 53.1 mm to the left and 6.0 mm anterior to the Cz of the standard international 10–20 system. The peak latency of the largest signal at channel (A) was 140 ms after the initiation of the finger movement. The localized large signal was also observed at the channel indicated as (B) in Figure 3; it was located 11.5 mm left and 15.1 mm anterior to Cz. The peak latency of the largest signal at channel (B) was 30 ms. Topographies of signal amplitude were obtained at selected time points: −1500 ms, −1000 ms, 30 ms, and 140 ms (Figure 4). The topography at a latency of −1500 ms shows that localized signals were observed in the vicinity of channel (B). This localized signal increased up to a peak latency of 30 ms in both channel (A) and channel (B). After the peak latency of 30 ms, the polarity of the signal in channels (A) and (B) changed abruptly, peaking at a latency of 140 ms. The topography at a peak latency of 140 ms indicated that the signals around channel (A) were locally spread over the posterior region. The topography change from −1600 ms to 500 ms across the scalp is shown in Appendix A.

In this study, we demonstrated the potential of our system to measure fast brain signals equivalent to those measured by EEG/MEG with excellent spatial resolution. Our technique enables the measurement of brain signals with high spatial resolution by biasing a magnetic field toward the brain as if it were a probe; thus, it was designated as a magnetically biased probe (MBP). Movement-related signals measured by EEG include the readiness potential (RP) or Bereitschaftspotential (BP), a gradual change in cortical potential that begins to develop approximately 2.0 to 1.5 s before the onset of voluntary movement [22,23]. In this study, signal changes corresponding to RP/BP were observed with our MBP system, beginning at approximately −2.0 s and peaking at latencies of 30 and 140 ms (Figure 3 and Figure 4). These results indicate that the MBP system can successfully measure brain signals with high temporal resolution, similar to that of EEG [24]. In addition, the signals in this study were observed before the onset of finger movement, indicating that these signals were not artifacts of finger movement. In EEG, the distribution of brain potentials is distorted by the inhomogeneity of the electrical conductivity of the head, resulting in a distribution that differs from the actual distribution of brain activity sites [6]. The largest amplitude of the movement-related potentials evoked by finger movement in EEG was measured mainly in the area around Cz [25]. The primary motor cortex (M1) is located in the posterior portion of the frontal lobe in the precentral gyrus. The area in M1 representing the hands and fingers is called the “hand knob” [26]. According to a previous MRI study, the location of the “hand knob” in the left hemisphere of the brain innervating the right body was reported to be between C1 (in the international 10–20 system) and C3h (in the international 10–5 system) [26]. The location of channel (A) in Figure 3, where the largest signal was measured by the MBP system, is consistent with the location of the “hand knob” reported in the MRI study. The supplementary motor area (SMA), which is located in the medial region of the frontal lobe anterior to the primary motor cortex, is also involved in finger movements [27]. Studies using functional MRI to measure brain signals during hand movements have reported activity in the SMA and the “hand knob” in the primary motor cortex [28]. The location of channel (B) in Figure 3, where a localized large signal was measured, is consistent with the location of the SMA [29]. After the onset of finger movement at a latency of 140 ms, the amplitude of the signal at channel (A) near the hand area of the primary motor cortex increased and spread to the posterior area. This suggests that afferent proprioceptive signals transmitted from the finger to the primary somatosensory cortex (S1), which is located posterior to the primary motor cortex, can be successfully observed using the MBP system with excellent spatial resolution [30].

The quality of the signals obtained by MBP was superior to that of EEG/MEG, as the signals shown in Figure 3 were obtained without the frequency filters required for EEG/MEG. MBP could measure cortical activity in a localized brain region just below the coil. On the other hand, the current 159-channel system has a limitation of missing brain activity in areas without coils. To measure brain activity throughout the entire head, coils should be placed densely on the scalp. The potential of MBP to accurately detect dynamic brain activity can be useful in a variety of applications, including cases in which conventional noninvasive measurement techniques of brain function are limited in their ability to diagnose brain diseases.

## 4. Conclusions

In this study, we demonstrated that the whole-head MBP system developed can be used to measure dynamic brain activity with excellent spatial accuracy. Movement-related brain signals, including signal changes corresponding to RP/BP, were measured at reasonable locations near the hand region of the M1 and SMA. Solving an inverse problem with no unique solution, as is the case with EEG/MEG, is not necessary for the MBP system as this system directly provides the spatial distribution of dynamic brain activity as topography on the head surface. Invasive brain–machine interfaces are used to obtain low-noise brain signals directly from the surface of the brain by inserting electrodes into the skull; however, the risk of infection or injury to brain tissue is a concern [31]. Because the magnetic field used in MBP is as strong as the geomagnetic field, the MBP technique is safe, noninvasive, and easy to use. Moreover, the whole-head MBP system is useful for diagnosing and studying higher brain functions, such as language, thought, memory, and sensory and motor brain functions, because the fast dynamics of the whole region of the cerebral cortex can be observed using the MBP system. The whole-head MBP system introduced in this study could reveal unclarified brain functions.

## Figures and Tables

**Figure 1 bioengineering-11-00917-f001:**
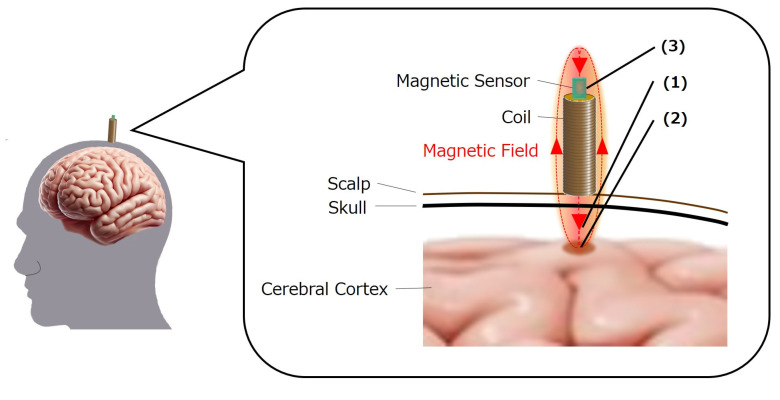
Measurement of neural signals in the cerebral cortex using a static magnetic field. (1) A static magnetic field generated by a coil on the scalp passes through the region of the cerebral cortex below the coil. (2) Static magnetic field fluctuates according to neural electromagnetic activity in the cerebral cortex through which the static magnetic field passes. (3) Magnetic sensor at the top of the coil measures neural activity in the cerebral cortex as a fluctuation in the magnetic field.

**Figure 2 bioengineering-11-00917-f002:**
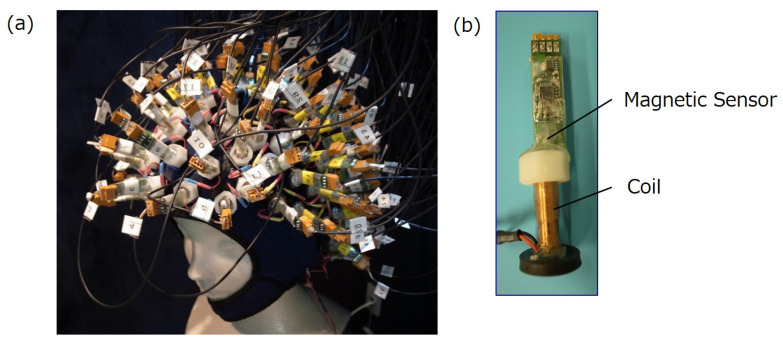
(**a**) Noninvasive whole-head system for measuring brain signals under static magnetic field bias. A total of 159 pairs of magnetic sensors and coils were placed on the scalp with a neoprene cap. (**b**) Each magnetic sensor was connected to a coil using a plastic nut.

**Figure 3 bioengineering-11-00917-f003:**
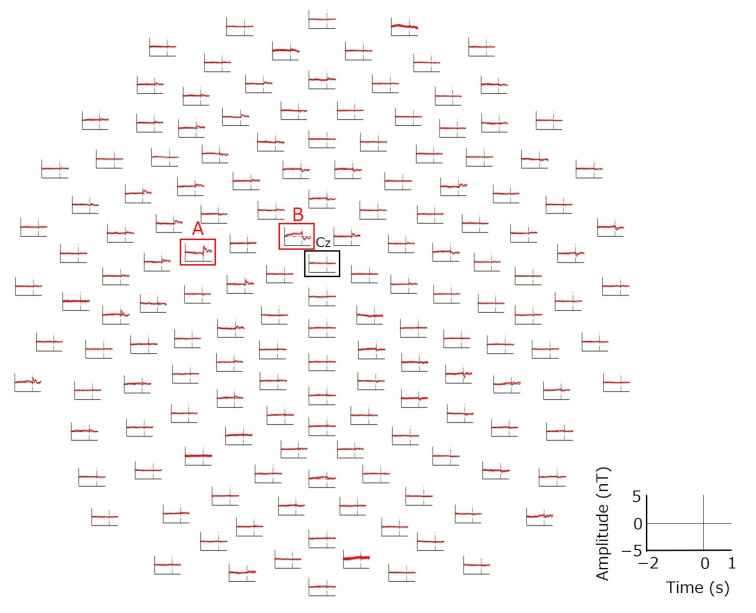
Grand averages of movement-related signals evoked by extension of the right index finger were measured in three participants using the developed whole-head MBP system. Signals for all 159 channels across the scalp surface are presented as if looking down on the scalp surface with the nose (anterior) at the top.

**Figure 4 bioengineering-11-00917-f004:**
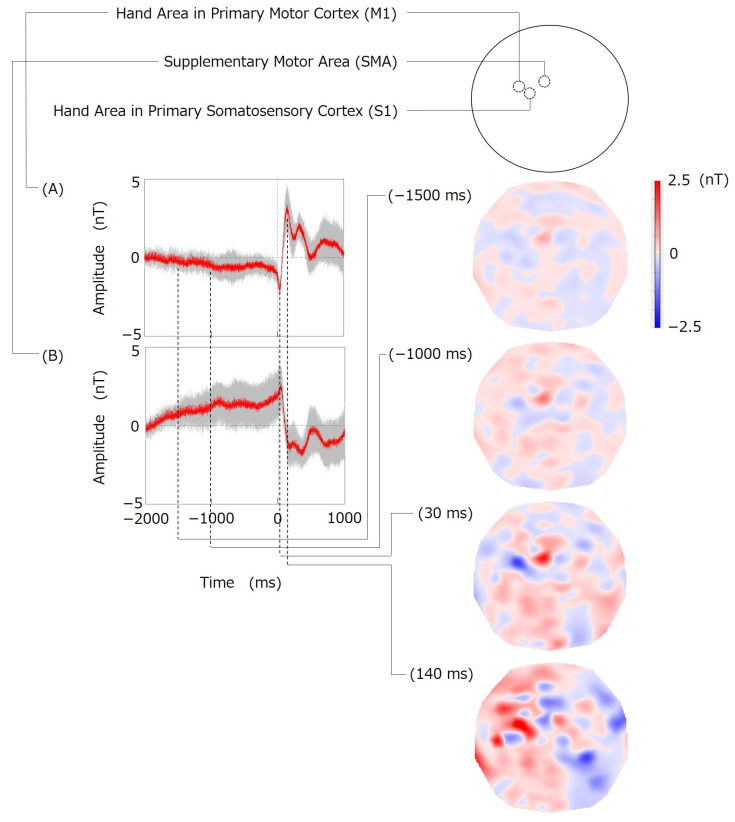
(**Left**) Movement-related signals of channels (A) and (B) in Figure 3. Gray lines indicate the standard deviation at each sampling time. (**Right**) Topographies of signal distributions across the scalp at selected time points: −1500 ms, −1000 ms, 30 ms, and 140 ms. Maps are presented as if looking down on the scalp surface with the nose (anterior) at the top. Positive and negative signals are shown in red and blue, respectively.

## Data Availability

The data that support the findings in this study are available from the corresponding author upon reasonable request.

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
