# Peer review of "Whole-Head Noninvasive Brain Signal Measurement System with High Temporal and Spatial Resolution Using Static Magnetic Field Bias to the Brain"

_bioengineering, 2024, doi:10.3390/bioengineering11090917_

Round 1
Reviewer 1 Report
Comments and Suggestions for Authors
The paper presents a novel noninvasive brain signal measurement technique that utilizes a biased static magnetic field. This technique aims to provide high temporal and spatial resolution for brain function measurement. The authors successfully demonstrate its efficacy by measuring movement-related brain signals. Considering the promising nature of the technique and the positive results, I recommend accepting the paper with major revisions. Some comments and questions below may be helpful for enhancing the quality of the work.
1. The authors should clarify certain aspects of the methodology and discuss potential limitations more comprehensively.
2. The experimental data presented in this work are relatively few. If possible, the authors are suggested to provide more characterization and experimental data for confirming the reliability of the referring technique.
3. The authors are suggested to discuss potential clinical applications of the MBP system and how it might contribute to the diagnosis and treatment of neurological disorders in the end of conclusions.
4. How does the MBP system handle artifacts and noise, such as those arising from muscle activity or external electromagnetic interference? A more in-depth explanation should be highlighted in the revised manuscript.
5. The authors should provide more information about the data analysis and signal processing techniques used in the study.
Comments on the Quality of English LanguageThe English language needs to be polished.
Author Response
【Response to Reviewer #1】
We sincerely appreciate the valuable comments on our manuscript entitled “Whole-Head Noninvasive Brain Signal Measurement System with High Temporal and Spatial Resolution Using Static Magnetic Field Bias to the Brain” We have substantially revised the manuscript. Our individual responses to each of the comments are listed as follows.
Comment 1:
The authors should clarify certain aspects of the methodology and discuss potential limitations more comprehensively.
Response:
We have added the explanation about the method of signal processing as follows. (p.3, line 113)
“The signals were averaged and topographic maps were drawn using MATLAB (MathWorks, USA).”
We have added the explanation about the limitation of the system introduced in the study as follows. (p.4, line 167)
“On the other hand, the current 159-channel system has a limitation of missing brain activity in areas without coils. To measure brain activity throughout the entire head, coils should be placed densely on the scalp.”
Comment 2:
The experimental data presented in this work are relatively few. If possible, the authors are suggested to provide more characterization and experimental data for confirming the reliability of the referring technique.
Response:
We have added the gray lines indicating the standard deviation in the data of Figure 4 to provide the reliability of the technique.
Comment 3:
The authors are suggested to discuss potential clinical applications of the MBP system and how it might contribute to the diagnosis and treatment of neurological disorders in the end of conclusions.
Response:
We have added the sentences at the end of the conclusion as follows. (p.4, line 184)
“Moreover, the whole-head MBP system is useful for diagnosing and studying higher brain functions, such as language, thought, memory, and sensory and motor brain functions, because the fast dynamics of the whole region of the cerebral cortex can be observed using the MBP system. The whole-head MBP system introduced in this study could reveal unclarified brain functions.”
Comment 4:
How does the MBP system handle artifacts and noise, such as those arising from muscle activity or external electromagnetic interference? A more in-depth explanation should be highlighted in the revised manuscript.
Response:
We have added the explanation about the quality of the signals measured by the system as follows. (p.4, line 165)
“The quality of the signals obtained by MBP was superior to that of EEG/MEG, as the signals shown in Fig. 3 were obtained without the frequency filters required for EEG/MEG.”
Comment 5:
The authors should provide more information about the data analysis and signal processing techniques used in the study.
Response:
We have the explanation that the data analysis was conducted with MATLAB as follows. (p.3 line 113)
“The signals were averaged and topographic maps were drawn using MATLAB (MathWorks, USA).”
Reviewer 2 Report
Comments and Suggestions for Authors
The paper calls: "Whole-Head Noninvasive Brain Signal Measurement System with High Temporal and Spatial Resolution Using Static Magnetic Field Bias to the Brain" and concerned of new approach of health monitoring system. The advantage of article original topic of research and adequate references list (24 ones).
The main question is: Why author don't tell about novelty or originality of the work? If this novelty work author should add "At the first time presented new method of brain monitoring using magnetic field..."
Author Response
【Response to Reviewer #2】
We sincerely appreciate the valuable comments on our manuscript entitled “Whole-Head Noninvasive Brain Signal Measurement System with High Temporal and Spatial Resolution Using Static Magnetic Field Bias to the Brain” We have substantially revised the manuscript. Our individual responses to each of the comments are listed as follows.
Comment 1:
The main question is: Why author don't tell about novelty or originality of the work? If this novelty work author should add "At the first time presented new method of brain monitoring using magnetic field..."
Response:
We have described that this technique is novel in the abstract and introduction as follows. (p.1 line 14) (p.2 line 63)
“We developed a novel noninvasive brain signal measurement technique with high temporal and spatial resolution by biasing a static magnetic field emitted from a coil on the head to the brain. In this study, we applied this technique to develop a groundbreaking system for noninvasive whole-head brain function measurement with high spatiotemporal resolution across the entire head.”
“In this study, we introduced a groundbreaking system for whole-head noninvasive measurement of brain signals with excellent temporal and spatial resolution using this technique. The system was validated by measuring movement-related signals evoked by volitional finger movements.”
Reviewer 3 Report
Comments and Suggestions for Authors
The authors developed a noninvasive brain signal measurement technique with high temporal and spatial resolution using a method of biasing a static magnetic field emitted from a coil on the head to the brain. However, there are many flaws in the manuscript that need to be revised rigorously. Below are the major comments:
1) The abstract must be strengthened by emphasizing the prime bounds meaningfully.
2) In keywords, please remove the numbering and restrict it to 5 words.
3) Please provide a schematic representation of the work for a better understanding of readers.
4) What are the hardware and software configuration? Data acquisition and analysis.
5) How the noninvasively picked-up brain signal is amplified and enhanced for analysis.
6) The introduction should be a more comprehensive and detailed discussion of the fundamentals of state-of-the-art needs including brain-machine interface, spatial resolution, and noninvasive signal measurement.
7) What is the novel significance of this proposed work?
8) Authors should consider adding new recently reported relevant articles to strengthen the proposed work: doi.org/10.1016/j.bspc.2024.106565; /doi.org/10.1016/j.cobme.2023.100505
9) Add the comparison table with recent state-of-the-art showing the parameters.
10) What is the key difference between the proposed work and published work: 10.1109/JTEHM.2020.3039043 Looks very similar. Please justify ad provide the statistics.
11) Fig 2 should be labeled as (a) and (b) with captions and internal labeling
12) Fig 3 looks too messy and unclear. It's hard to understand
13) Fig 4 should be more elaborated
14) Please discuss the challenges and limitations.
15) Add the appendix of abbreviation at the end.
16) Proper recent references should be cited and remove the older or irrelevant ones. Add more references.
17) Poor English language, I recommend checking the language with native English
Comments on the Quality of English LanguageThe authors developed a noninvasive brain signal measurement technique with high temporal and spatial resolution using a method of biasing a static magnetic field emitted from a coil on the head to the brain. However, there are many flaws in the manuscript that need to be revised rigorously. Below are the major comments:
1) The abstract must be strengthened by emphasizing the prime bounds meaningfully.
2) In keywords, please remove the numbering and restrict it to 5 words.
3) Please provide a schematic representation of the work for a better understanding of readers.
4) What are the hardware and software configuration? Data acquisition and analysis.
5) How the noninvasively picked-up brain signal is amplified and enhanced for analysis.
6) The introduction should be a more comprehensive and detailed discussion of the fundamentals of state-of-the-art needs including brain-machine interface, spatial resolution, and noninvasive signal measurement.
7) What is the novel significance of this proposed work?
8) Authors should consider adding new recently reported relevant articles to strengthen the proposed work: doi.org/10.1016/j.bspc.2024.106565; /doi.org/10.1016/j.cobme.2023.100505
9) Add the comparison table with recent state-of-the-art showing the parameters.
10) What is the key difference between the proposed work and published work: 10.1109/JTEHM.2020.3039043 Looks very similar. Please justify ad provide the statistics.
11) Fig 2 should be labeled as (a) and (b) with captions and internal labeling
12) Fig 3 looks too messy and unclear. It's hard to understand
13) Fig 4 should be more elaborated
14) Please discuss the challenges and limitations.
15) Add the appendix of abbreviation at the end.
16) Proper recent references should be cited and remove the older or irrelevant ones. Add more references.
17) Poor English language, I recommend checking the language with native English
Author Response
【Response to Reviewer #3】
We sincerely appreciate the valuable comments on our manuscript entitled “Whole-Head Noninvasive Brain Signal Measurement System with High Temporal and Spatial Resolution Using Static Magnetic Field Bias to the Brain” We have substantially revised the manuscript. Our individual responses to each of the comments are listed as follows.
Comment 1:
The abstract must be strengthened by emphasizing the prime bounds meaningfully.
Response:
We have revised the abstract to emphasize the prime bounds as follows. (p.1 line 16)
“In this study, we applied this technique to develop a groundbreaking system for noninvasive whole-head brain function measurement with high spatiotemporal resolution across the entire head.”
Comment 2:
In keywords, please remove the numbering and restrict it to 5 words.
Response:
We have removed the numbering and restricted it to 4 words. (p.1, line 22)
Comment 3:
Please provide a schematic representation of the work for a better understanding of readers.
Response:
We have revised Figure 1 to show a schematic representation of the work.
Comment 4:
What are the hardware and software configuration? Data acquisition and analysis.
Response:
We have deserved the hardware and software configuration as follows. (p.2, line 93)
“The signal outputs of the magnetic sensors were acquired using data acquisition modules (NI-9205, National Instruments, USA) inserted into slots in a chassis (cDAQ-9179, National Instruments, USA). Signals from the 159 magnetic sensors were simultaneously sent to the PC through the USB port of the cDAQ-9179 and acquired by a system developed by LabVIEW (National Instruments, USA).”
We have the explanation that the data analysis was conducted with MATLAB as follows. (p.3 line 113)
“The signals were averaged and topographic maps were drawn using MATLAB (MathWorks, USA).”
Comment 5:
How the noninvasively picked-up brain signal is amplified and enhanced for analysis.
Response:
We did not amplify the measured brain signals as described in p.2, line 93.
Comment 6:
The introduction should be a more comprehensive and detailed discussion of the fundamentals of state-of-the-art needs including brain-machine interface, spatial resolution, and noninvasive signal measurement.
Response:
We have added the explanation about the fundamentals of state-of-the-art needs including brain-machine interface, spatial resolution, and noninvasive signal measurement as follows. (p.2, line 59)
“Brain–machine interfaces (BMI), which use brain signals to operate computers and devices, require dynamic and spatially accurate brain information. However, conventional techniques to measure brain function do not provide sufficient information for BMI.”
Comment 7:
What is the novel significance of this proposed work?
Response:
We have added the explanation of the significance of this study as follows. (p.2, line 63)
“We developed a novel noninvasive brain signal measurement technique with high temporal and spatial resolution using a method that biases a static magnetic field emitted from a coil on the head to the brain. In this study, we introduced a groundbreaking system for whole-head noninvasive measurement of brain signals with excellent temporal and spatial resolution using this technique. The system was validated by measuring movement-related signals evoked by volitional finger movements.”
Comment 8:
Authors should consider adding new recently reported relevant articles to strengthen the proposed work: doi.org/10.1016/j.bspc.2024.106565; /doi.org/10.1016/j.cobme.2023.100505
Response:
doi.org/10.1016/j.bspc.2024.106565; (A hyperdimensional framework: Unveiling the interplay of RBP and GSN within CNNs for ultra-precise brain tumor classification) is a study about classification of brain tumors.
/doi.org/10.1016/j.cobme.2023.100505; (Non-invasive brain imaging to advance the understanding of human balance) is a study about EEG measurement of balance control.
We have carefully considered these studies, but they seem not directly relevant to this study.
Comment 9:
Add the comparison table with recent state-of-the-art showing the parameters.
Response:
We described the comparison with the conventional techniques of noninvasive brain measurement in the introduction section. We added the sentence that our system can measure fast brain signals equivalent to those measured by EEG/MEG with excellent spatial resolution as follows. (p.3, line 130)
“In this study, we demonstrated the potential of our system to measure fast brain signals equivalent to those measured by EEG/MEG with excellent spatial resolution.”
Comment 10:
What is the key difference between the proposed work and published work: 10.1109/JTEHM.2020.3039043 Looks very similar. Please justify ad provide the statistics.
Response:
We have described that this study is the development of a whole-head noninvasive system to measure brain signals with excellent temporal and spatial resolution using the technique of the published work of 10.1109/JTEHM.2020.3039043 as follows. (p.2, line 65)
“In this study, we introduced a whole-head noninvasive system to measure brain signals with excellent temporal and spatial resolution using this technique. The system was validated by measuring movement-related signals evoked by volitional finger movements.”
Comment 11:
Fig 2 should be labeled as (a) and (b) with captions and internal labeling
Response:
We labeled (a) and (b) in Fig.2.
Comment 12:
Fig 3 looks too messy and unclear. It's hard to understand
Response:
We submitted clearer image of Fig.3 separately.
Comment 13:
Fig 4 should be more elaborated
Response:
We have revised Fig.4 more elaborated.
We added the topography changed from -1600 ms to 500 ms across the scalp as Fig. S1. (p.3 line 129)
Comment 14:
Please discuss the challenges and limitations.
Response:
We have added the explanation about the challenges and limitation of the system introduced in the study as follows. (p.4, line 167)
“On the other hand, the current 159-channel system has a limitation of missing brain activity in areas without coils. To measure brain activity throughout the entire head, coils should be placed densely on the scalp.”
Comment 15:
Add the appendix of abbreviation at the end.
Response:
We looked carefully for words, but could not find any abbreviations that needed explanation.
Comment 16:
Proper recent references should be cited and remove the older or irrelevant ones. Add more references.
Response:
We added recent references: 8, 11, 14, 16, 18, 23.
Comment 17:
Poor English language, I recommend checking the language with native English.
Response:
This paper has been edited for English language, grammar, punctuation, and spelling by Enago, the editing brand of Crimson Interactive Pvt. Ltd under Normal Editing.
Round 2
Reviewer 1 Report
Comments and Suggestions for Authors
The author has made the necessary revisions. The manuscript is suggested for publication now.
Comments on the Quality of English LanguageThe English language is fine.
Reviewer 3 Report
Comments and Suggestions for Authors
The author has made significant changes to the revised version of manuscript and it can be accepted in its current form for publication
Comments on the Quality of English LanguageThe author has made significant changes to the revised version of the manuscript and it can be accepted in its current form for publication